# Robotic-Assisted Rehabilitation for Post-Stroke Shoulder Pain: A Systematic Review

**DOI:** 10.3390/s23198239

**Published:** 2023-10-03

**Authors:** Rossana Gnasso, Stefano Palermi, Antonio Picone, Domiziano Tarantino, Giampiero Fusco, Maria Michelina Messina, Felice Sirico

**Affiliations:** Public Health Department, University of Napoli “Federico II”, Via Pansini 5, 80131 Naples, Italy; rossanagns@yahoo.it (R.G.); antonio.picone@unina.it (A.P.); domiziano22@gmail.com (D.T.); giampiero.fusco@libero.it (G.F.); mariamichelinamessina@gmail.com (M.M.M.); sirico.felice@gmail.com (F.S.)

**Keywords:** shoulder pain, robotics, hemiplegia, stroke rehabilitation, systematic review

## Abstract

Post-stroke shoulder pain (PSSP) is a debilitating consequence of hemiplegia, often hindering rehabilitation efforts and further limiting motor recovery. With the advent of robotic-assisted therapies in neurorehabilitation, there is potential for innovative interventions for PSSP. This study systematically reviewed the current literature to determine the effectiveness of robotic-assisted rehabilitation in addressing PSSP in stroke patients. A comprehensive search of databases was conducted, targeting articles published up to August 2023. Studies were included if they investigated the impact of robotic-assisted rehabilitation on PSSP. The outcome of interest was pain reduction. The risk of bias was assessed using the Cochrane database. Of the 187 initially identified articles, 3 studies met the inclusion criteria, encompassing 174 patients. The reviewed studies indicated a potential benefit of robotic-assisted rehabilitation in reducing PSSP, with some studies also noting improvements in the range of motion and overall motor function. However, the results varied across studies, with some showing more significant benefits than others, because these use different protocols and robotic equipment.

## 1. Introduction

Stroke remains one of the most arduous health challenges worldwide, and it is a primary cause of mortality and long-term disability [1]. The aftermath of a stroke can be multifaceted, affecting an individual’s ability to perform daily activities, engage in social interactions, and maintain a satisfactory quality of life. One of the most pervasive and debilitating motor impairments following a stroke is hemiplegia [1]. This condition, characterized by paralysis on one side of the body, often manifests in both the upper and lower limbs. This leads to a cascade of complications, including reduced voluntary movement, increased muscle tone, and restricted passive range of motion (ROM).

A particularly distressing consequence of this muscular imbalance in the upper limb is the onset of post-stroke shoulder pain (PSSP) syndrome [2]. PSSP not only exacerbates the physical discomfort of stroke survivors but also poses significant psychological challenges. Persistent pain and discomfort can hinder rehabilitation efforts, demotivate patients from participating in therapeutic activities, and, in some cases, further impede motor recovery [3]. This underscores the urgency of finding effective interventions to manage and alleviate PSSP.

Over the years, the medical community has explored a myriad of interventions to address PSSP [4]. These have spanned from oral pharmacological treatments, which target both nociceptive and neuropathic pain pathways, to more physical interventions such as the use of orthoses, specialized physiotherapy sessions, and targeted injections including steroids or botulinum toxin [5]. Each of these approaches has its merits, but none has emerged as a definitive solution.

A promising frontier in stroke rehabilitation is the field of robotic-assisted therapies. These interventions, which leverage advancements in technology and biomechanics, enhance motor and cognitive recovery in stroke patients [6]. The integration of technology allows for personalized therapy, real-time feedback, and the potential for ramified rehabilitation exercises, thereby enhancing patient engagement [7]. Moreover, these new technologies have already been tested for the rehabilitation of the shoulder, indicating how this complex articulation can benefit from such a new approach [8,9,10]. 

While the primary focus of robotic-assisted rehabilitation has traditionally been on motor function restoration, its potential therapeutic benefits for conditions such as PSSP are beginning to gain attention. Given the increasing prevalence of stroke and the persistent challenge of PSSP, it is imperative to explore all avenues of intervention.

In this context, our study embarks on a mission to systematically review the existing literature. Our goal was to determine the effectiveness and potential of robotic-assisted rehabilitation as a viable intervention for PSSP in stroke patients.

## 2. Materials and Methods

### 2.1. Data Sources and Searches

This study adhered to a standard systematic review procedure, following the guidelines outlined in the Cochrane Handbook and the recommendations detailed in the PRISMA statement [11]. The investigation involved searching several databases for articles published up to August 2023, including the Cochrane Central Register of Controlled Trials (CENTRAL), EMBASE, MEDLINE, PEDro, PubMed, and CINAHL, according to specific syntax rules. In addition, a manual search of published studies was performed, and the reference lists of the retrieved studies were reviewed. The search strategy combined terms such as “robotic,” “rehabilitation,” “robotic-assisted rehabilitation,” “hemiplegic shoulder pain,” “post-stroke shoulder pain,” and “pain” using Boolean operators. 

### 2.2. Study Selection

To ensure the reliability of the evidence, the analysis was limited to randomized controlled trials, excluding observational studies, case reports, and other studies with lower methodological rigor. Only articles in English were considered. 

Studies were considered eligible according to the following PICO model: (P) Participants: Individuals with post-stroke shoulder pain (PSSP).(I) Intervention: Robotic-assisted physiotherapy. There were no limitations on the specific robotic equipment used. Any intervention that employed robotic-assisted physiotherapy to address PSSP was deemed eligible.(C) Comparison of conventional physiotherapy: This encompassed kinesiotherapy involving passive, assistive, and active exercises, as well as stretching exercises, and included studies reporting conventional physical therapy techniques such as hot pack application, electrical stimulation, kinesiotaping, and non-robotic approaches. The authors opted to include randomized trials comparing conventional physiotherapy with robotic-assisted physiotherapy, where the latter group exclusively or in combination with conventional physiotherapy received robotic-assisted treatment.(O) Outcome: Pain assessment scales. The included studies were required to report at least one pain assessment scale. If multiple scales for pain were presented within a single study, all scores were incorporated into the analysis. Composite assessment scales that combined various elements related to function, strength, and pain into a singular score were only considered if it was feasible to extract only the pain domain assessment value.

Studies that (1) enrolled stroke patients lacking PSSP, (2) excluded robotic-assisted physiotherapy in at least one arm of the study design, (3) included pediatric or adolescent stroke cases, (4) solely reported outcomes of function, disability, quality of life (QoL), or other factors (excluding pain assessment), (5) involved animal models, (6) were reported in languages other than English, and (7) lacked full-text or numerical data were excluded from consideration.

After eliminating duplicates, the titles and abstracts of the articles identified in the literature search were independently reviewed by two researchers (R.G. and S.P.) to determine their suitability for analysis. In the event of disagreements, a third reviewer (F.S.) was consulted to reach a consensus.

### 2.3. Data Extraction and Assessment of the Risk of Bias

The full text of the possibly eligible research papers was collected and evaluated separately by the two authors (R.G. and S.P.). In the event of any disagreements between the authors regarding the eligibility of a particular study, they resolved the matter by discussing it with the other researchers. A form for data extraction was employed to gather pertinent details from the selected studies, enabling the assessment of the risk of bias and synthesis of evidence. This form encompassed the following: authors, journal, publication year, sample size, study methodology, types of interventions (details of the experimental and control interventions), assessed outcomes, and durations of follow-up. The studies included in the analysis were evaluated for potential bias using Cochrane tools, specifically the revised tool for assessing the risk of bias in randomized trials (RoB 2) [12]. Various potential sources of bias were considered, including the randomization process, deviations from intended interventions, incomplete outcome data, outcome measurement, and selection of reported results. Following the Cochrane guidelines, each aspect was categorized as “low risk,” “high risk,” or “some concerns” regarding bias. The risk of bias was assessed by two authors (A.P. and F.S.) independently, and disagreements were resolved by discussion with other researchers. Due to the subjective nature of pain perception assessed through self-reported scales and the impracticality of blinding in the group receiving robotic-assisted physiotherapy (both participants and intervention providers were aware of the treatment being administered), the unblinding was considered as the most influential element affecting the risk of bias in the study.

## 3. Results

### 3.1. Search Results

From a pool of 187 initially identified articles, once duplicates were removed, the titles and abstracts of 120 studies were screened. Following the criteria for inclusion and exclusion as mentioned earlier, 16 studies were chosen for an in-depth assessment, and their complete texts were obtained. Upon a thorough examination of these full texts, 13 of the 16 studies were disqualified for subsequent reasons. 

Paolucci et al. [13] were excluded from consideration in this review because their study involved stroke patients in a randomized design with two arms: robotic-assisted physiotherapy and robot-assisted physiotherapy combined with botulinum toxin. While pain was an outcome examined, the primary focus of the intervention was the addition of botulinum toxin to the rehabilitative treatment. Both groups received the same robotic-assisted treatment.

Masiero et al. [14] conducted a study to investigate the effectiveness of robot-assisted physiotherapy in stroke patients. Although the study design aligned with our review, the included stroke patients did not experience PSSP. The authors assessed pain outcomes and found a similar occurrence of PSSP in both groups (two patients in each group). However, the main outcome was not related to the impact of robotic-assisted physiotherapy on PSSP.

Calabrò et al. [15] and Park et al. [16] conducted non-controlled trials with stroke patients, focusing on restoring motor function and not reporting pain outcomes. Similarly, Carpinella et al. [17] evaluated the effects of robot therapy on upper body kinematics and arm function in post-stroke patients. They investigated the addition of robotic treatment to conventional physiotherapy and found better results in patients who received the robotic intervention. 

Taravati et al. [18] conducted a similar study using the ReoGo™-Motorika upper extremity rehabilitation system. They demonstrated comparable outcomes between groups, suggesting that adding robotic-assisted intervention to conventional physiotherapy could be effective for neurological rehabilitation in stroke patients. Nevertheless, as in studies reported previously, pain was not considered as an outcome, and PSSP treatment did not represent the focus of the study. 

Some studies identified through a systematic research of literature have a three-arm design. Yuan et al. [19] conducted a study comparing conventional training, unilateral robotic training, and bilateral robotic training. They concluded that clinical advantages and electrophysiological improvements were observed only in the bilateral robot-assisted training group in addition to conventional treatment.

Simkins et al. [20] conducted a study on robotic unilateral and bilateral upper limb movement training for stroke survivors with chronic hemiparesis. Their three-arm study evaluated the therapeutic efficacy of a dual-arm exoskeleton system. Pain outcome was graphically represented, showing no differences among the groups. However, patients with PSSP were not detailed in the study.

Takibahashi et al. (2022) [21] investigated robot-assisted training for upper-limb hemiplegia in chronic stroke using three arms: conventional self-training plus conventional therapy, robotic self-training (ReoGo-J) plus conventional therapy, and robotic self-training plus constraint-induced movement therapy. They found no significant differences among the groups.

These studies adopted a study design that fit this systematic review; however, they did not include patients with PSSP. Takebahashi et al. (2011) [22] assessed the differences in applying CIMT or CIMT combined with robotic-assisted physiotherapy in stroke patients. However, even their focus was on motor improvement and excluded patients with PSSP.

Jang et al. [23] compared two different applications of functional electrical stimulation for shoulder subluxation in patients with stroke: normal FES vs. brain–computer interface-controlled FES. Both approaches resulted in reduced VAS scores without significant differences between the groups.

Dohle et al. [24] reported a pilot study investigating the effect of robotic-assisted physiotherapy on glenohumeral joint subluxation in patients with chronic stroke. Their uncontrolled single-arm study reported positive results in terms of subluxation reduction, functional outcome, spasticity reduction, and muscle strength improvement. Pain measures were not included.

Irina et al. [25] published an abstract presented at the 9th World Congress of the International Society of Physical and Rehabilitation Medicine. The study included 24 subjects with unilateral neglect and shoulder subluxation who were randomized into two groups. Both groups received conventional physiotherapy, and the intervention group also received Armeo^®^ treatment. However, the authors did not provide quantitative data on improvement or pain outcomes. As a result, only three studies were considered eligible for this systematic review. The outcomes of the study search and selection are reported in Figure 1.

### 3.2. Characteristics of the Included Studies

The trials included in the analysis enrolled 174 patients. The characteristics of the included studies are reported in Table 1.

Aprile et al. [26] aimed to determine the prevalence of PSSP and its relationship with upper limb recovery. This study was a secondary analysis of a multicenter randomized controlled trial published previously by the authors [29]. This secondary analysis was conducted to assess the pain outcome related to PSSP in stroke patients. The study involved 224 patients who underwent either conventional or robotic rehabilitation. Only patients suffering from PSSP were included, and pain data were collected at T1 (after 30 sessions of treatment) and T2 (after three months from the end of treatments) in 122 patients. The intervention group included patients treated with a set of robotic and sensor-based devices (Motore, Humanware, and Amedeo, Diego e Pablo from Tyromotion). The control group received physiotherapy treatment that focused on sensory stimulation, stretching, passive mobilization, functional training, and task practice. Both groups received interventional therapy daily for 45 min, five days a week, for 30 sessions. All patients received additional conventional physiotherapy (six times/week) for 45 min, which focused on lower limbs, sitting and standing training, balance, and walking. During this treatment, the therapists were instructed not to provide additional upper limb therapy. Therefore, patients differed between groups only in terms of treatment directed to upper limb rehabilitation (conventional physiotherapy versus robot-assisted physiotherapy). The intensity of shoulder pain was assessed using the Numerical Rating Scale (NRS) and Douleur Neuropathique 4 (DN4). Scores of NRS and DN4 were categorized on the basis of specific cut-offs: NRS 1–4 as mild pain and NRSS 5–10 as moderate–severe pain. Among DN4, patients scoring up to 4 were categorized as DN4+, whereas others as DN4. The results showed that 28.9% of the patients reported moderate/severe shoulder pain, with 19.6% showing a neuropathic component. The incidence of PSSP was inversely correlated with upper limb motor function and directly correlated with the time since stroke. Pain intensity was higher in women and in patients with neglect syndrome. No differences in PSSP were reported between the groups, demonstrating that both conventional and robotic rehabilitation led to a reduction in pain, and this improvement was maintained three months after rehabilitation. This study highlights the importance of considering PSSP in rehabilitation planning to optimize recovery outcomes and mainly focuses on the identification of risk factors involved in the onset of PSSP. No adverse effects were reported in either group, and the authors specified that the loss of patients during follow-up was not related to the adverse effects of dissatisfaction with the type of treatment received.

Kim et al. [27] found that robotic-assisted shoulder rehabilitation therapy effectively reduced hemiplegic shoulder pain and self-reported shoulder-related disability. The authors demonstrated the effectiveness of a newly developed robot for performing joint mobilization and stretching exercises with patients lying in the supine position. They assessed 59 patients and enrolled 38 in the trial. Two patients were lost at follow-up (one in each arm of treatment) for reasons unrelated to the treatment. Both groups received conventional physiotherapy through passive exercise to improve the range of motion and rehabilitation based on the Bobath approach. The authors reported some other interventions, such as hot packs, ultrasound, transcutaneous electrical nerve stimulation, and analgesics applied equally between groups. The intervention group received additional robotic-assisted therapy for 30 min per day, 5 times per week for 4 weeks (20 sessions). The robotic equipment adopted in this study was a prototype that could perform robotic-assisted physiotherapy in the supine position and admit passive movements and stretching exclusively on abduction planes. The outcomes were assessed at baseline, after 20 sessions of intervention, and at the 4-week follow-up. Significant improvements were observed in pain levels, passive range of motion, and shoulder disability questionnaire scores in the intervention group compared with the control group, with improvements in VAS score, passive range of motion degrees in abduction movement of the affected upper limb, and Korean version of the Shoulder Disability Questionnaire.; these improvements were sustained at the 4-week follow-up. Among the measures of outcome, the authors included an ultrasonographic assessment of the affected shoulder, categorizing some features such as effusion of the biceps tendon, tendinosis of the supraspinatus, subacromial–subdeltoid bursitis, and partial or full thickness tear of the rotator cuff. However, no significant differences were found between the groups. The authors assessed the presence of glenohumeral subluxation as a confounder at baseline between groups and reported this condition in 61% of patients in both groups, but did not consider the effects of treatments on subluxation as an outcome. The authors clearly declare no adverse effects in either group, demonstrating the safety of the equipment and protocol adopted.

Serrezuela et al. [28] assessed 18 patients and enrolled 16 with PSSP. They conducted a pilot study comparing robotic-assisted physiotherapy with conventional physiotherapy. The interventional group used an exoskeleton with 4 degrees of freedom to perform antigravitational movements in a seated position. Patients in this group performed movements in several planes, such as abduction of the shoulder, flexion and extension of the shoulder and elbow, external and internal rotation of the shoulder, and pronosupination of the wrist. They received robotic treatment for 1 h daily in 5 weekly sessions for 3 months. No other kind of therapy was administered in this group. Patients allocated to the conventional physiotherapy group received superficial thermotherapy, lymphatic massage, and kinesiotherapy, performing the same movements as those of patients in robotic-assisted physiotherapy. This latter intervention was carried out in 1 h sections daily, in 5 weekly sessions, and according to the Kabat approach. The authors assessed several outcomes. Pain was assessed using a 4-degree ordinal scale: severe (0), moderate (1), light (2), and normal (3). Spasticity was evaluated according to the Modified Ashworth Scale and muscular strength was assessed according to the Medical Research Council Scale. The active range of motion of the shoulder was categorized as “non-functional shoulder” (active flexion less than 60° and active abduction less than 40°), “with improvement” (active flexion less than 60° and active abduction less than 40°), and “functional shoulder” (active flexion more than 90° and active abduction more than 75°). Moreover, the authors assessed the degree of subjective general satisfaction using a Likert scale from 1 (Nothing) to 10 (A lot), and humeral scapular subluxation was classified as “absent”, “moderate” (less than 7mm), and “severe” (more than 7mm). The authors concluded that robotic-assisted physiotherapy could significantly reduce pain compared with conventional physiotherapy. Moreover, reporting patients’ single data in each group, the authors showed that pain decreases faster by adopting exoskeleton treatment, and this result is maintained during follow-up assessments. Moreover, outcomes regarding spasticity, muscular strength, and active range of motion were significantly improved in the robot-assisted physiotherapy group compared with the conventional physiotherapy group. Specifically, the active range of motion was improved in abduction, flexion, and rotations. No adverse effects were reported in either group, and high levels of satisfaction were reported in both groups (with higher scores in the robotic-assisted physiotherapy group). Regarding glenohumeral subluxation, the authors reported its clinical and radiological absence in 88% of the whole sample, showing that this condition is not the only determinant in patients with PSSP. Nevertheless, this condition improved in the robotic-assisted physiotherapy group. 

Concerning the risk of bias assessment of the included studies, the results are graphically reported in Figure 2. All included studies showed some minor concerns about the domain “deviation from the intended intervention” according to the tool adopted, mainly related to blinding procedures issues. Aprile et al. [26] reported that patients were randomly assigned to the robotic group or the control group. Details about the randomization process are specified in the authors’ main analysis [29], where they reported a solid procedure based on a randomization sequence generated using the R (version 3.3.0, R Core Team, Vienna, Austria) package blockrand, with random block sizes ranging from 2 to 8. Randomization was stratified according to disease onset and age, and the list was prepared by an investigator with no clinical role in the study. Kim et al. [27] reported a randomization procedure using a computer-generated randomization sequence created using the block randomization of 2. Serrezuela et al. [28] reported a randomization procedure performed by a physiotherapist unrelated to the study using a random number table. In all studies, the personnel performing randomization were blinded to the patients’ conditions. 

While the included studies declared that assessors were blinded to patients’ treatment, participants and operators cannot be blinded to the enrollment in the robotic or control group of a specific subject. All included studies adopted validated measures of outcomes previously used in the literature. No studies were declared to have recorded missing outcome data that can bias the reported results, and all pre-specified analysis plans have been reported to avoid the selection of reported results.

## 4. Discussion

The aim of our systematic review was to evaluate the effectiveness of robotic-assisted rehabilitation for treating PSSP. Following a thorough screening process, only three studies were included in our analysis, totaling 174 patients. These studies provided evidence that robotic-assisted therapy exhibited significant improvements in reducing PSSP symptoms, enhancing joint mobility, and promoting overall motor recovery compared with conventional rehabilitation methods.

The present study attempts to systematically synthesize evidence regarding the influence of PSSP on upper limb recovery after rehabilitation, which appears to be a relatively unexplored area of research. While the existing literature supports the effectiveness of robot-assisted training in improving outcomes for stroke patients, it should be noted that none of these studies specifically address post-stroke shoulder pain or its outcome after robotic-based rehabilitation. Therefore, this study aims to fill this knowledge gap and shed light on the potential benefits of robotic-assisted therapy in addressing PSSP and enhancing overall recovery outcomes. Our findings align with the broader literature on the subject. Bertani et al. [30], in their meta-analysis, showed that robot-assisted training was effective in improving outcomes in stroke patients. These findings are also shared by the systematic reviews of Verbeek et al. [31] and Kwakkel et al. [32]. These studies, among others, underscore the potential of robotic-assisted therapy in addressing post-stroke pain and improving overall recovery outcomes. 

The available evidence suggests that incorporating robotic-assisted rehabilitation into the treatment of PSSP can yield significant benefits. The precise and consistent nature of robot-assisted interventions may contribute to a more targeted and effective approach to pain management. However, it is important to acknowledge that one study [26] did not find notable differences between the robotic and conventional groups in terms of pain reduction. This indicates that while robotic-assisted physiotherapy offers certain advantages, traditional physiotherapy remains an effective option for addressing PSSP. 

Secondary outcomes, including improvements in glenohumeral subluxation measurements, reduced spasticity, and increased passive range of motion without pain, were also observed in the group undergoing robotic-assisted rehabilitation. These additional positive results are important because they are closely linked to alleviation of pain in patients with PSSP [28]. The advancements seen in these secondary outcomes indicate that robotic-assisted rehabilitation may provide a more effective approach to address the root causes of PSSP than traditional physiotherapy methods.

The included studies have some differences in their study design. Two out of the three studies [26,28] investigated robotic-assisted physiotherapy versus conventional treatment. In contrast, Kim et al. [27] used robotic-assisted physiotherapy in addition to conventional therapy. This difference in the study design could have affected the results. Robotic-assisted physiotherapy represents an intervention option for complex clinical entities such as PSSP. Several rehabilitative interventions can be performed on these patients, and they are widely discussed in the literature [5]. Therefore, even if a direct comparison of robotic-assisted versus conventional treatment could be effective from a research point of view, combining robotic treatment with conventional treatment represents the most valuable scenario in clinical practice. The positive results shown in adopting both robotic and conventional treatments is encouraging for treating PSSP. 

No included studies attempted to correlate results with the etiology of PSSP. Kim et al. [27] reported the incidence of glenohumeral subluxation in both intervention groups as baseline characteristics, without assessing this condition after treatment. Serrezuela et al. [28] demonstrated the reduction in glenohumeral subluxation after robotic treatment, but this result was limited to a single subject in both groups. Only Aprile et al. [26] attempted to categorize patients according to different components of pain by assessing the prevalence of neuropathic pain in stroke patients affected by PSSP, demonstrating that a small percentage of patients in both groups have PSSP associated with positive findings of neuropathic pain. Nevertheless, these patients obtained similar results with robotic or conventional physiotherapy. Moreover, Kim et al. were unable to correlate PSSP with ultrasonographic findings. These considerations underline the complex etiology of PSSP, defining it as a difficult multifactorial entity. 

Aprile et al. [26] demonstrated that the time since stroke and associated features such as hemineglect could impact PSSP, making it even more difficult to establish the right time and modality of intervention in these stroke patients. 

Another point of discussion is related to the robotic equipment used in the selected studies. Different robotic equipment was used by Aprile et al. [26] in a multicentric study. Kim et al. [27] tested a prototype performing treatment in the supine position, and Serrezuela et al. [28] tested an exoskeleton in the seated position. Moreover, some of this equipment only allows for passive movements in nongravitational planes, whereas others allow for active-assisted movement in antigravitational models. Therefore, the heterogeneity of interventions, strictly related to the differences in the adopted equipment, could have an impact on the effectiveness of rehabilitative results. However, this aspect of robotic rehabilitation is difficult to assess and makes it harder to conduct comparable studies across different rehabilitative structures. In the evolving landscape of robotic rehabilitation, the diverse utilization of various robots and protocols underscores the potential of future advancements [33]. To harness this potential, it is paramount to foster interdisciplinary collaborations between rehabilitation professionals and engineers, bridging clinical needs with technical innovations. Research should pivot towards demonstrating the clinical efficacy of robot-assisted rehabilitation, particularly its influence on neuroplasticity, using advanced imaging and neurophysiological methods. The usability of robots can be significantly enhanced by refining the human–robot interface, emphasizing wearability and daily life applicability, and integrating cutting-edge technologies like virtual reality, augmented reality, and artificial intelligence. Clear technical specifications are essential to facilitate result comparisons across studies, encompassing treatment parameters like volume and frequency. The development of sophisticated software tools for data analysis related to human–robot interactions and patient motor performance evaluations is equally crucial. Emphasis on optimizing available prototypes, as highlighted by Gandolfi et al. [33], can expedite international market certifications, with a focus on technical characteristics comparisons, such as exoskeletons versus end-effector robots. The future beckons the creation of adaptable exoskeletons tailored to user morphology and anthropometry, and software that discerns voluntary intentions on cognitive and physical fronts, making robots more embodied and intuitive.

Other variables that can impact the outcomes are related to the characteristics of the intervention, such as volume, frequency, and intensity. All included studies performed treatment through daily sessions of 1 h, and no adverse effects have been reported. Movements were limited to a predetermined pain-free range of motion. Some previous studies have reported an increased risk of damage and worsening of pain in PSSP patients treated with excessive exercise protocols [34]. 

Our findings underscore the potential efficacy of this type of rehabilitation in patients with complex conditions.

Our review is notable for its meticulous methodology, which ensured the inclusion of only top-tier studies. This rigorous approach lends significant credibility to our findings and reinforces their validity. Notably, our review stands out by emphasizing the long-term advantages offered by robotic-assisted therapy, a perspective overlooked in other studies. However, it is important to acknowledge certain limitations of our study. One major limitation lies in the relatively small number of included studies that exhibited some concerns about the risk of bias as determined through the Cochrane’s assessment tool. 

Particularly in rehabilitative research (Aprile et al. [26]), it is impossible to design protocols for studies that ensure blinding of participants and personnel, given that therapies in rehabilitation are often active and thus difficult to “hide” for both the experimenter and the subject. This also occurs in robotic rehabilitation treatments. We are confident that with the ongoing research process and new emerging technologies such as artificial intelligence, we could adopt sham therapies in future studies in the field of rehabilitation. Nevertheless, solid randomization procedures blinding of the assessors, and reporting of all pre-specified analyses appear to be valid procedures that limit bias in the studies. 

In addition, we must recognize that excluding non-English articles may have resulted in overlooking relevant research. Moreover, it becomes obvious that there was considerable variation among the protocols used in robot rehabilitation across the included studies, differing both with respect to duration and style. Therefore, caution should be exercised when interpreting these results because of the potential confounding factors associated with such heterogeneity.

Finally, another critical aspect worth noting is the limited availability of high-quality research meeting our criteria on this topic overall, highlighting a clear necessity for further investigation within this domain.

Given the promising results of robotic-assisted therapy, future research should delve deeper into understanding the mechanisms underlying its efficacy. It would be beneficial to explore the specific components of robotic therapy that contribute the most to pain reduction and motor recovery. In addition, long-term studies spanning years, rather than months, could provide insights into the sustained benefits and potential drawbacks of this approach. Finally, as technology continues to evolve, evaluating the effectiveness of newer robotic rehabilitation devices will be crucial.

## 5. Conclusions

Available data on the effectiveness of robotic-assisted rehabilitation as a treatment option aimed at reducing pain in stroke patients with PSSP are limited. The included studies reported different equipment and protocols regarding volume, intensity, and the type of treatment. Nevertheless, robotic-assisted rehabilitation is not harmful in these patients and appears to reduce pain. The application of robotic-assisted physiotherapy to stroke patients with PSSP seems reasonable, even if further studies are necessary to compare different robotic equipment and different protocols of intervention and to establish which treatment has more impact on pain control in PSSP, according to other clinical characteristics of the patients. These data could then be used as a basis for further investigation aimed at assessing the application of robot-based rehabilitation for pain management in stroke patients with PSSP. The future beckons the creation of adaptable exoskeletons tailored to user morphology and anthropometry, and software that discerns voluntary intentions on cognitive and physical fronts, making robots more embodied and intuitive.

## Figures and Tables

**Figure 1 sensors-23-08239-f001:**
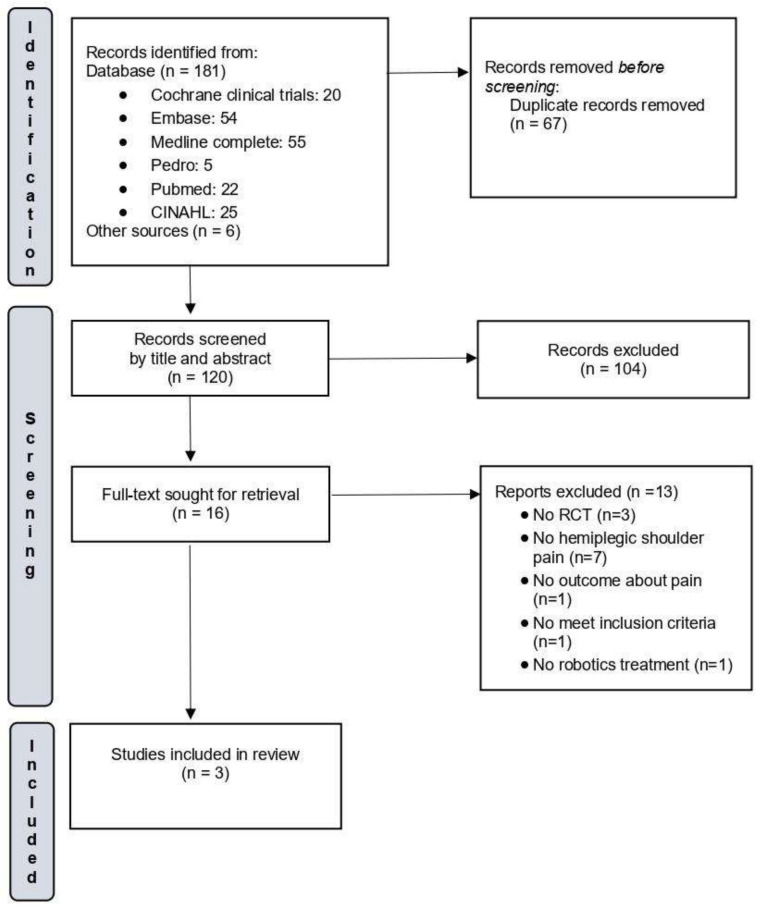
Flow diagram of the selected studies.

**Figure 2 sensors-23-08239-f002:**
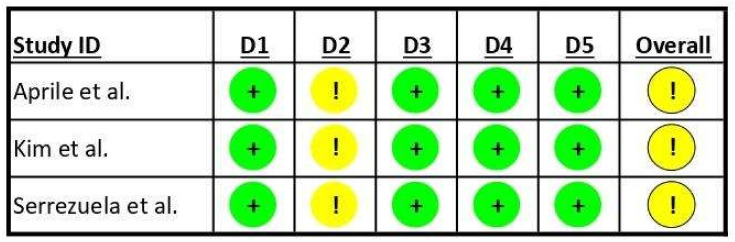
Risk of bias in the included studies. “+” Low risk of bias; “!” Some concerns about risk of bias; D1: randomization process; D2: deviations from intended interventions; D3: missing outcome data; D4: measurement of outcome; D5: selection of reported result; Overall: overall risk of bias. Aprile et al., 2021 [26]; Kim et al., 2019 [27]; Serrezuela et al., 2020 [28].

**Table 1 sensors-23-08239-t001:** Characteristics of the included studies.

Authors	Year	Study Design	Sample (m/f)	Intervention	Robotic Equipment	Pain-Related Outcome	Follow-Up
Aprile et al. [26]	2021	RCT (Robotic-assisted physiotherapy versus conventional physiotherapy)	224 (127/97) randomized 122 completed the follow-up assessment	- Conventional treatment *n* = 64) The treatment focused on sensory stimulation, stretching, passive mobilization, functional training, and task practice. - Robotic treatment (*n* = 58) The treatment was performed daily for 45 min, five days a week, for 30 sessions in both groups.	A set of robotic and sensor-based devices (Motore, Humanware; and Amadeo, Diego, and Pablo, from Tyromotion)	- Numerical Rating Scale (NRS) A score between 1 and 4 was categorized as “mild pain/influence” from 5 to 6 as “moderate pain/influence” and a score equal to or higher than 7 as “severe pain/influence” - Douleur Neuropathique 4 (DN4) to diagnose neuropathic pain. A score of 4 indicates a neuropathic origin of pain.	Patients were evaluated at baseline (T0), after conventional or robotic rehabilitation treatment (T1), and 3 months after the end of treatment (T2).
Kim et al. [27]	2019	RCT (Robotic-assisted physiotherapy + conventional physiotherapy versus conventional physiotherapy)	38 (22/14) randomized 36 completed the follow-up assessment	- Conventional treatment (*n* = 18) Conventional physical therapy directed at both improving upper extremity mechanics through PROM exercises and reducing neurologic injury based on the Bobath approach - Robotic-assisted shoulder rehabilitation therapy (RSRT) (*n* = 18) Conventional physical therapy + robotic-assisted shoulder rehabilitation therapy administered for 30 min per day, 5 times per week, for 20 sessions for 4 weeks.	Prototype robot performing mobilization and stretching exercises on the shoulder while the patient is lying in the supine position	10-point VAS Pain-free PROM of the shoulder Korean version of the Shoulder Disability Questionnaire (K-SDQ)	Baseline (T0), Immediately after the intervention (4 weeks) (T1), At the 4-week follow-up)
Serrezuela et al. [28]	2020	RCT (Robotic-assisted physiotherapy versus conventional physiotherapy)	16 (9/7) randomized 16 completed the follow-up assessment	Conventional therapy group (*n* = 8) Superficial thermotherapy (infrared) as analgesic, antispasmodic, and anti-inflammatory treatment and as preparation for exercise, lymphatic massage, and kinesitherapy (1 h) including manual exercises based on Kabat approaches - Robotic-assisted shoulder rehabilitation therapy (*n* = 8) Therapy was performed in a sitting position. The following movement routines were selected for the study: flexion/extension of the shoulder and elbow, external and internal rotation and abduction of the upper limb, and prono/supination of the wrist. 1 h per day, 5 weekly sessions with antigravitational movements.	Exoskeleton in a grounded robotic platform of 4 freedom degrees	The severity degree of the painful shoulder was defined in four grades: - Severe (0), pain, and functional limitation while resting with all movements limited. - Moderate (1), pain that intensifies with movement and is very light while resting, the movements are painful, but there is no limitation in the joint range. - Light (2) without pain at rest only occurs with rapid movements or under active mobilization - Almost normal (3), where only pain appears or limitation to active movements is resisted.	Baseline (T0), Every 10 sessions of treatment, with a monthly summary At 3 months (T1) Single data subjects were collected and data at baseline and at 3 months are reported in the study.

## Data Availability

Not applicable.

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
