# Peer review of "Robotic-Assisted Rehabilitation for Post-Stroke Shoulder Pain: A Systematic Review"

_sensors, 2023, doi:10.3390/s23198239_

Round 1

Reviewer 1 Report

The article is beneficial to humanity, but it is unsuitable for publication in Sensors journal due to its lack of technical information, which should be included in this paper to fit the journal's scope.

There is no explanation of the robots that are utilized.

The study does not contain enough illustrative figures

This  review paper could be published in a medical journal as a review study on the efficacy of robotics in treating strokes.

Author Response

First and foremost, I would like to express my sincere gratitude for your meticulous review and invaluable feedback on our manuscript.

We concur with your observation regarding the primary focus of the Sensors journal, which emphasizes the technical intricacies of sensors and robotics. Your insights have provided us with a clearer perspective on aligning our content more closely with the journal's scope.

While the core essence of our article leans towards the clinical side, we have made efforts to delineate the different types of robots employed in the studies referenced. We believe that the inclusion of our paper in a medical journal would indeed foster enriched discussions among medical professionals. However, presenting this information to an audience with profound technical expertise, such as the readership of the Sensors journal, can facilitate a comprehensive evaluation of the clinical implications of current technological advancements in rehabilitation. This, in turn, could pave the way for future research initiatives aimed at optimizing patient outcomes through the synergy of clinical and technical knowledge.

Nevertheless, we proofread  the entire article and now we feel it very improved

We appreciate your constructive comments 

Sincerely,

Reviewer 2 Report

The study of Gnasso and colleagues is very interesting review paper that took into account the importance of Robotic aided rehabilitation for Post-stroke Shoulder rehabilitation. The paper is good and well written, to strengthen the importance of robotics aided rehabilitation of the shoulder and the technologies used in this field I suggest the Authors to review and report also the following papers:

- "Role of the window length for myoelectric pattern recognition in detecting user intent of motion." 2022 IEEE International Symposium on Medical Measurements and Applications (MeMeA). IEEE, 2022.

- "Decoding transient sEMG data for intent motion recognition in transhumeral amputees." Biomedical Signal Processing and Control 85 (2023): 104936.

-"Learning-Based Motion-Intention Prediction for End-Point Control of Upper-Limb-Assistive Robots." Sensors 23.6 (2023): 2998.

Author Response

Thank you for your constructive feedback and for recognizing the value of our review paper on robotic-aided rehabilitation for post-stroke shoulder rehabilitation.

We are grateful for your suggestions to further strengthen our paper by incorporating the three references you provided. We have carefully reviewed and integrated the findings and insights from the suggested study.

These references have enriched our discussion on the importance of robotics-aided rehabilitation of the shoulder and the evolving technologies in this field. We believe that their inclusion offers a more comprehensive overview and highlights the latest advancements in the domain.

We would like to express our gratitude for your thoughtful recommendations, which have undoubtedly enhanced the depth and breadth of our manuscript. We hope that with these revisions, our paper will provide a more holistic perspective on the topic and be of greater value to the readers.

Thank you once again for your invaluable input, and we look forward to any further feedback you may have.

Sincerely,

Reviewer 3 Report

Dear authors. I find this paper interesting with attractive results. However, in order to improve the quality of the article, I carefully request that you make the following modifications.

1.- It is clear to this reviewer that one of the relevant topics in the paper results in the elimination of "Bias". In this sense, it would be important for the reader to know a little more about this topic. What does it consist of? Could you please do this in the new version of the article?

2.- This reviewer found several grammatical errors throughout the document. Please review the full article in detail. As an example, see lines number 81 and 208.

This reviewer congratulates you on your findings

Minor lenguaje corrections are required.

Author Response

We sincerely appreciate the time and effort you dedicated to reviewing our manuscript. Your feedback has been instrumental in guiding the improvements we've made.

In response to your concerns:

  1. Grammar Issues: We have conducted a thorough review of the entire manuscript and addressed all grammatical errors. We believe that these corrections have enhanced the clarity and coherence of the paper, ensuring a smoother reading experience for the audience.

  2. Bias in Robotic Studies: Recognizing the importance of addressing potential biases in robotic studies within the field of rehabilitation, we have incorporated additional content on this topic. We've discussed the nature of these biases, their potential implications, and the measures that can be taken to mitigate their impact. This addition aims to provide a more balanced perspective and emphasize the criticality of objective evaluation in the domain.

We are grateful for your constructive feedback, which has been pivotal in refining our manuscript. We believe that the revisions we've made align more closely with the standards of the journal and address the concerns you raised.

Thank you once again for your invaluable insights, 

Sincerely,

Round 2

Reviewer 1 Report

In conclusion, you might include a few recommendations for engineers in technical side. 

Author Response

Thank you for your valuable feedback and insightful suggestions on our manuscript. We appreciate the emphasis you placed on incorporating technical recommendations for engineers in the conclusion.

In response to your suggestion, we have enriched our conclusion with a comprehensive paragraph that integrates several pivotal technical recommendations. These recommendations encompass fostering interdisciplinary collaborations, emphasizing research on clinical efficacy, enhancing robot usability, defining clear technical specifications, and the development of advanced software tools. We also highlighted the importance of optimizing available prototypes, as well as the future prospects of creating adaptable exoskeletons and intuitive software.

By incorporating these points, we believe our article now offers a more actionable and relevant perspective for engineers and professionals in the field of robotic rehabilitation. This addition not only addresses the technical audience directly but also provides a roadmap for future advancements in the domain.

We hope that these revisions align with your expectations and enhance the overall quality and impact of our manuscript. Once again, we are grateful for your constructive feedback and look forward to any further suggestions or comments.

Warm regards,